# Comprehensive Analysis of Human Subtelomeres by Whole Genome Mapping

**Eleanor Young**[1], **Heba Z. Abid**[1], **Pui-Yan Kwok**[2,3,4], **Harold Riethman**[5]\*, **Ming Xiao**[1,6]\*

**1** School of Biomedical Engineering, Drexel University, Philadelphia, PA, United States of America, **2** Cardiovascular Research Institute, University of California–San Francisco, San Francisco, CA, United States of America, **3** Department of Dermatology, University of California–San Francisco, San Francisco, CA, United States of America, **4** Institute for Human Genetics, University of California–San Francisco, San Francisco, CA, United States of America, **5** Medical Diagnostic & Translational Sciences, Old Dominium University, Norfolk, VA, United States of America, **6** Institute of Molecular Medicine and Infectious Disease in the School of Medicine, Drexel University, Philadelphia, PA, United States of America

\* hriethma@odu.edu HR; Ming.Xiao@drexel.edu MX

**Data Availability Statement:** All optical mapping files are available from NCBI BioProject database under BioProject PRJNA418343. (URL: https://www.ncbi.nlm.nih.gov/bioproject/418343).

## Abstract

Detailed comprehensive knowledge of the structures of individual long-range telomere-terminal haplotypes are needed to understand their impact on telomere function, and to delineate the population structure and evolution of subtelomere regions. However, the abundance of large evolutionarily recent segmental duplications and high levels of large structural variations have complicated both the mapping and sequence characterization of human subtelomere regions. Here, we use high throughput optical mapping of large single DNA molecules in nanochannel arrays for 154 human genomes from 26 populations to present a comprehensive look at human subtelomere structure and variation. The results catalog many novel long-range subtelomere haplotypes and determine the frequencies and contexts of specific subtelomeric duplicons on each chromosome arm, helping to clarify the currently ambiguous nature of many specific subtelomere structures as represented in the current reference sequence (HG38). The organization and content of some duplicons in subtelomeres appear to show both chromosome arm and population-specific trends. Based upon these trends we estimate a timeline for the spread of these duplication blocks.

## Author summary

The ends of human chromosomes have caps called telomeres that are essential. These telomeres are influenced by the portions of DNA next to them, a region known as the subtelomere. We need to better understand the subtelomeric region to understand how it impacts the telomeres. This subtelomeric region is not well described in the current references. This is due to large variations in this region and portions that are repeated many times, making current sequencing technologies struggle to capture these regions. Many of these variations are evolutionary recent. Here we use 154 different samples from the 26 geographic regions of the world to gain a better understanding of the variation in these

**Funding:** This study was funded by grants from the US National Institutes of Health (https://www.nih.gov/). The grant numbers are: R01HG005946 (MX and PYK), R21CA177395 (MX and HR), and R01CA140652 (HR). The funders had no role in study design, data collection and analysis, decision to publish, or preparation of the manuscript.

**Competing interests:** The authors have declared that no competing interests exist.

regions. We found many new haplotypes and clarified the haplotypes existing in the current reference. We then examined population and chromosome specific trends.

## Introduction

Telomere-adjacent DNA helps regulate telomere (TTAGGG)n tract lengths and telomere integrity. A family of long noncoding telomeric repeat-containing RNA (TERRA) molecules is transcribed from subtelomeres into (TTAGGG)n tracts [1–3], and association of TERRA with other shelterin components and telomeric DNA is necessary for telomere integrity and function [1, 4, 5]. Subtelomeric DNA elements cis to (TTAGGG)n tracts regulate both TERRA levels and haplotype-specific (TTAGGG)n tract lengths and stabilities [4, 6–9] with evidence for epigenetic modulation of these effects [9–12]. Extended subtelomere regions contain both coding and non-coding transcripts, the abundance and regulation of which are likely to depend upon the specific haplotypes and copy number of the DNA encoding them. Some of these transcripts such as those encoding human WASH proteins are clearly functional, but most are not well-characterized [13–17]. *De novo* deletion of specific subtelomeric duplications can cause disease in some contexts [18]. Long-range interactions of telomeres with functional subtelomeric genes has been observed, with the expression of these genes sometimes regulated in a telomere length-dependent fashion [19, 20].

Large structural variations occur frequently in subtelomeric DNA, often associated with loss or gain of large pieces of evolutionarily recent segmental duplications. Ambiguities in sequence localization because of segmental duplication content, as well as the presence of alternative haplotypes differing by relatively large insertions, deletions, and more complex sequence organization differences, have contributed to gaps and misassemblies in subtelomeric regions of the human reference sequence. In the current version (HG38) single haplotypes of many subtelomeres have been sequenced to the beginning of terminal repeat (TTAGGG)n tracts, whereas others still contain (TTAGGG)n-adjacent gaps. These ambiguities are represented in HG38 as strings of unknown nucleotides ("NNN's") intended to represent stretches of DNA with the number of "NN's" corresponding to estimated basepair (bp) length of the gap from the existing subtelomeric reference to the end of each respective chromosome arm [21, 22].

These ambiguities in the reference sequence, along with knowledge gained from limited long-range mapping studies that many additional large subtelomeric structural variations likely remain to be characterized [23], make the routine use of subtelomeric reference sequences problematic. Stong et al. [24] updated subtelomeric assemblies by using telomere clones from a fosmid structural variation resource[25] to fill in relatively small (TTAGGG)n-adjacent sequence gaps in the clone-based reference sequence, and re-defined subtelomere coordinates that exclude the (TTAGGG)n tract itself in order to create a custom subtelomere reference assembly ("Stong Assembly") useful for characterizing subtelomeric segmental duplications and extending the subtelomere paralogy blocks originally defined by Trask and co-workers [26]. This resulted in an assembly significantly more useful for subtelomere characterization and short-read sequence mapping purposes than previous ones; subtelomeres were operationally defined as the distal 500 kb regions of each chromosome arm, and encompassed all known multi-telomere segmental duplications (Subtelomere Repeat Elements, SREs). The SREs comprise roughly 25% of the entire subtelomere region and 80% of the most distal 100 kb of these regions; each defined subtelomere region also contained a stretch of 1-copy subtelomere-specific DNA on its centromeric end that definitively connects it with the

rest of the reference sequence [24]. The improved subtelomere assemblies are still subject to ambiguities associated with a few remaining large telomere-adjacent gaps as well as many very large structural variations that define alternative long-range haplotypes for an unknown number of individual subtelomeres in human populations.

We have previously used single-molecule optical mapping to identify long-range haplotypes in human genomes [27, 28], and showed that long (average 300kb) molecules mapped using this procedure can span segmental duplications in subtelomeres and connect chromosome ends to 1-copy arm-specific DNA [22]. Here we extend our analyses using high throughput optical mapping of large single DNA molecules in nanochannel arrays for 154 human genomes from 26 populations to present a comprehensive look at human subtelomere structure and variation. The results catalog many novel long-range subtelomere haplotypes and determine the frequencies and contexts of specific subtelomeric duplicons on each chromosome arm, helping to clarify the currently ambiguous nature of many specific subtelomeres as represented in the current reference sequence (HG38).

## Results

### Individual subtelomeric consensus maps containing large SRE regions

Subtelomeric repeat element (SRE) regions are located in the most distal stretches of human subtelomeres. Long SRE regions of about 300 kb have been identified in some alleles of the 1p, 8p and 11p telomeres, whereas 7 telomeres have minimal or no SRE content [17, 24, 29, 30]. Most SRE regions are 40–150 kb in size [24]. Physical linkage of 1-copy regions with telomeres on single large DNA molecules capable of spanning SRE regions is required for assembling individual subtelomeric consensus maps. Recently-developed high-throughput single-molecule genome mapping methods are well-suited for this challenge. In this method, genomic DNA is labeled at sites recognized by a sequence motif-specific nicking endonuclease, long genomic DNA fragments are isolated and imaged in nanochannel arrays to a high depth of coverage and contigs of these large genomic DNA fragments are assembled from these data. In our case, these maps are then compared with in silico-generated maps of subtelomeric reference sequences. Fig 1 shows the consensus map (yellow) and constituent single-molecules (brown) of the 3q subtelomere from the GM191025 genome aligned with HG38 reference (blue) and the 3q assembly from Stong et al. (2014; top). The paralog blocks of SREs are shown in the colored rectangles defined in Stong et al. (2014). The long DNA molecules shown here are at least 0.35Mb, reaching into the single copy region of the 3q arm. Their good alignment in the single copy region indicates these molecules belong to 3q. These molecules also contain 130 kb of extra sequences beyond the end of the incomplete HG38 reference and Stong et al. assembly; the nicking pattern of this extra sequence is consistent with paralogy blocks 1–5 as shown (dashed boxes on top of the molecules in Fig 1). The GM191025 genome map indicates there is 130 kb of DNA extending beyond the end of the HG38 reference sequence (teal arrow), including 60 kb accounted for by the gap sequence (black arrow). All of this additional DNA is associated with previously-identified SRE paralogy blocks.

### Discovery of novel subtelomeric structural variants, resolution of sequence gaps and delineation of long-range subtelomeric haplotypes across 154 genomes

To gain better insight into the subtelomeric regions, we next analyzed genome maps of subtelomeric regions of 154 human genomes selected from the 1000 genomes project [31]. These genomes include 3 males and 3 females from each of the 26 ethnic regions of the world. By

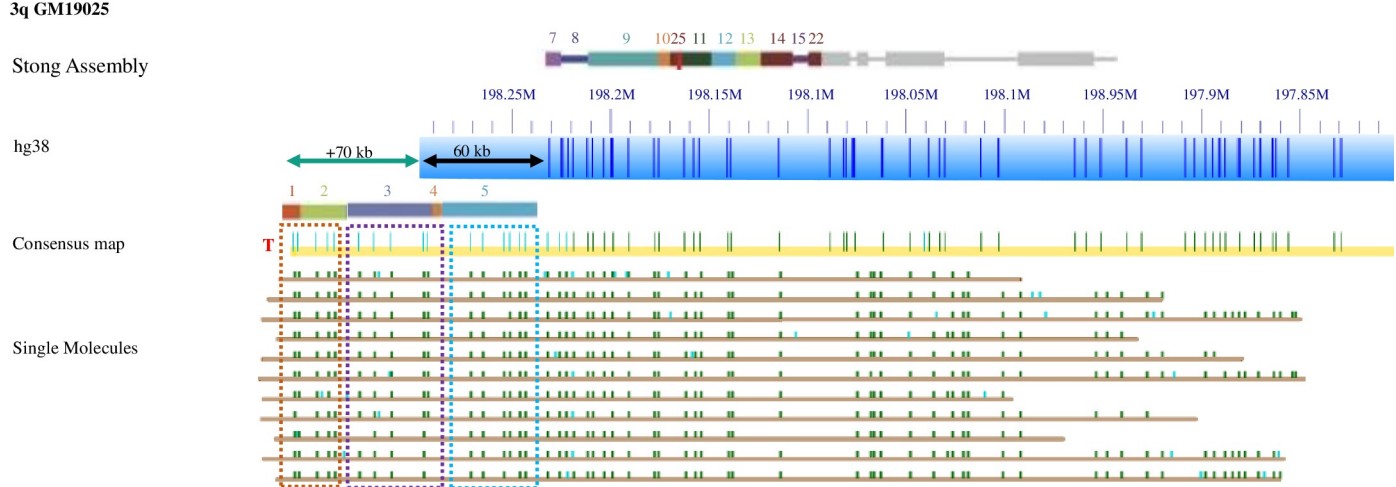

**Fig 1. Extended haplotypes in subtelomeric regions of 3q in GM191025 supported by single molecule evidence.** Colored rectangles represent paralogy blocks defined in the subtelomere assemblies of Stong et al. [24]. The blue bar shows the hg38 reference with Nt.BsPQ1 nick sites as dark blue dashes along it. The yellow bar shows the consensus contig for this sample, with dark green marks indicating a match to the reference and lighter green/blue showing nick sites without a reference match. The colored rectangles about the yellow bar show the paralogy blocks that match the pattern seen in the extended region. The brown rows indicate single molecules, which extend well past the block regions and into the single copy region. A teal arrow shows the distance, 70kb, from the telomere as defined by the Bionano single-molecule maps to the end of the HG38 reference assembly. A black arrow represents 60 kb of unknown sequence currently in the HG38 reference as 'N', an estimate of gap size to the end of the chromosome. Dashed boxes on top of the molecules indicate portions of the extended region that match to paralogy blocks 1–5 but are not in the current references for 3q. A red T indicates the telomeric end of the 3q map.

using this large and diverse sample set, we hope to form a more accurate representation of haplotypes and variations found in the subtelomeric regions of all chromosome arms.

**Highly variable subtelomeres.** Out of 46 chromosome arms, 18 (1p, 2q, 3q, 5q, 6p, 6q, 7p, 7q, 8p, 9p, 9q, 11p, 14q, 15q, 16q, 17q, 19p, 20p) are classified as highly variable. These are arms where structurally variant haplotypes were found in more than 10% of the total genomes analyzed [22]. Fig 2 summarizes the distribution of haplotypes for some of these highly variable chromosome arms. The remaining arms can be found in S1, S2, S3 and S4 Figs. The consensus maps of these subtelomeric regions show a wide range of variation between genomes, most strikingly in the length and sequence content of telomere-adjacent DNA segments. In many cases (1p, 6p, 7p, 9q, 11p, 16q, 19q, 20p), the HG38 reference doesn't represent the main haplotype. Fig 2 follows the same convention as in Fig 1, except the black dashed arrow signifies that a region of telomere-adjacent gap sequence in the HG38 reference should be deleted. A red 'T' indicates the Stong assembly reached the telomere sequences for that arm.

**1p**: For 96 genomes the chromosome starts near the 0.6Mb coordinate of the HG38 reference chromosome 1p arm, and for another 17 genomes it starts around 0.5Mb of the hg38 reference. The remaining 41 genomes failed to assemble. There are no perfectly aligned contigs between 0–0.5Mb of the HG38 reference. We confirmed this using an alternative labeling method, direct label enzyme (DLE; 36). In a separate study of 6 genomes, we tagged the telomeres with CRISPR-cas9, and found that all molecules in these genomes containing telomeres align to the hg38 reference starting at 0.6Mb [31, 32]. Fig 3 contains an example of one of these telomere images. This indicates that the first 500kb in hg38 reference for chromosome 1p contains incorrectly mapped DNA; indeed, analysis of DNA from each of the sequenced clones from this 500 kb region indicate that it is comprised entirely of segmental duplications, explaining its incorrect mapping location.

**2q**: For 2q, there is a major haplotype with 110 genomes that matches well to the hg38 reference. 20 genomes have an extension 45 kb more than the hg38 reference end. This is very

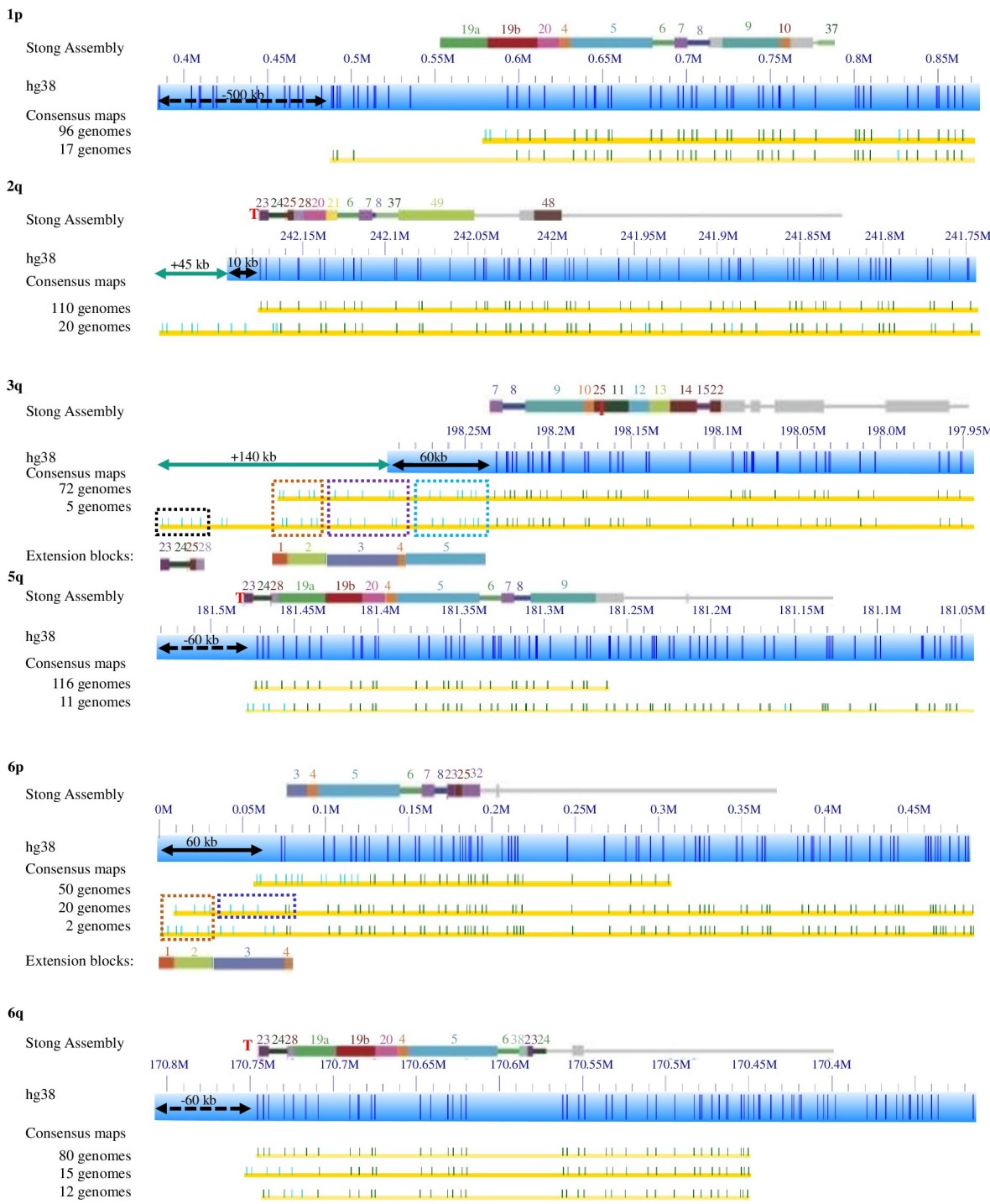

**Fig 2. Major haplotypes of highly variable subtelomere regions.** The Stong et al. assembly blocks are shown as colored rectangles above blue Bionano genome mapping bars. Yellow rows with green ticks show haplotypes below these. A teal arrow indicates the size of additional extended regions not covered by the reference. A black arrow represents unknown sequence currently in the HG38 reference as 'N', an estimate of gap size to the end of the chromosome. If the black arrow is dashed it signifies a region of unknown telomere-adjacent gap sequence that should be deleted. A red T indicates the Stong 2014 assembly reached the telomere, and the lack of one means that assembly was unable to reach the telomere repeats. Highly variable arms 1p, 2q, 3q, 5q, 6p, and 6q are included here. Additional highly variable arms (7p, 7q, 8p, 9p, 9q, 11p, 14q, 15q, 16q, 17q, 19p, 20p) can be found in S1 Fig through S4 Fig.

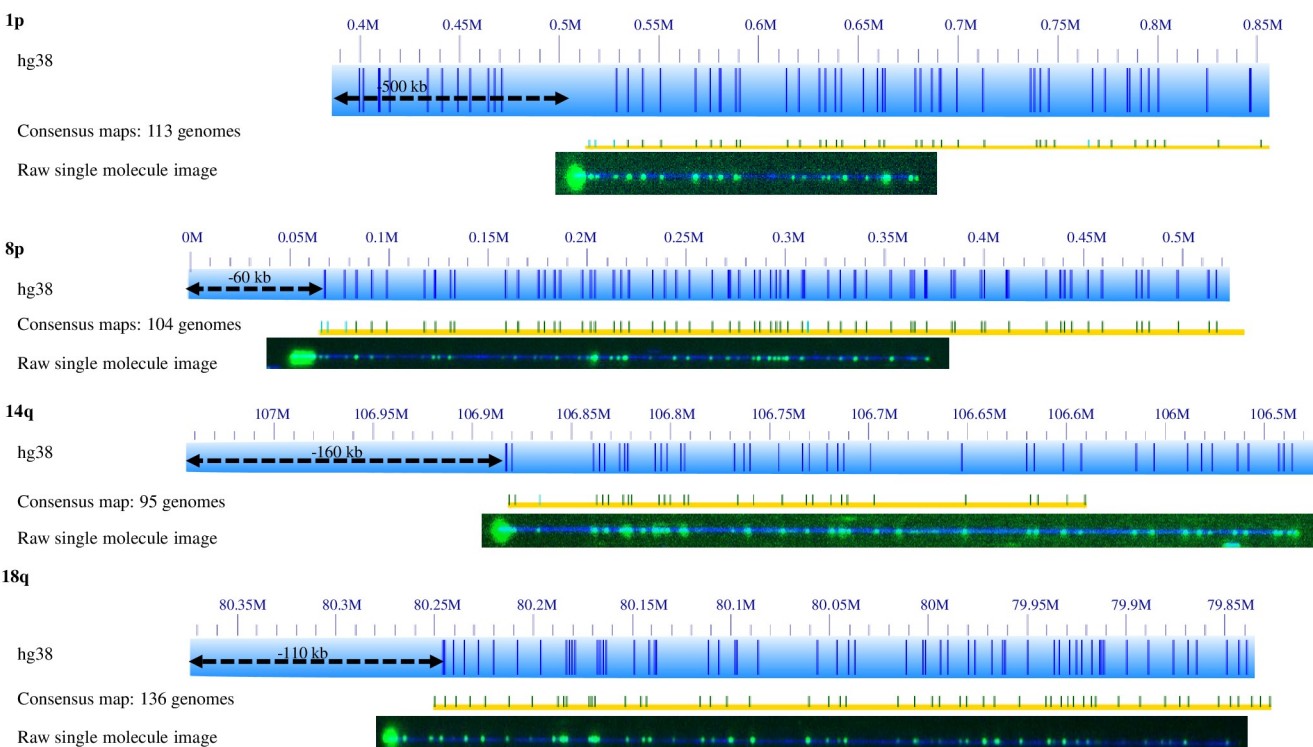

**Fig 3. Telomere labeling shows inaccurate sizing of telomere-adjacent gap segments in HG38 subtelomere regions.** Blue bars represent the nick sites in hg38 reference. Yellow bars with green Nt.BsPQ1 nick sites represent the main haplotype seen in the genomes. A black dashed arrow indicated the width of the telomere-adjacent gap sequence that should be deleted from the hg38 reference. An image below the haplotype shows a single telomere labeled molecule confirming the end of the chromosome arm. These telomeres were labeled using CRISPR-Cas9 to tag the telomere repeat and incorporate a fluorophore[32]. None of the subtelomeric haplotypes for each of these arms extends past the telomere label shown here.

similar in size to a 50 kb polymorphism at 2q detected independently using RARE cleavage mapping [33]. 3 genomes start at 242.12Mb, and DNA from the remaining 21 genomes failed to assemble a contig at this telomere.

**3q**: 72 genomes extend 130 kb beyond the Stong assembly and hg38 reference as shown for GM191025 (Fig 1). The genome map pattern of this region indicates that these genomes contain paralogy blocks 1–5, which are lacking in both the hg38 reference and the Stong Assembly (4). 5 genomes extend an additional 70 kb and contains paralogy blocks 23, 24, 25 and 28. Another 10 genomes contain variable extensions beyond the reference, and 23 genomes start at 198.16 Mb. DNA from 44 genomes failed to assemble a contig at this telomere.

**5q**: 135 genomes start at 181.48 Mb. 116 out of the 135 genomes match the hg38 reference and 19 genomes of 135 genomes have alternative patterns in the last 30kb adjacent to telomere. An additional 15 genomes start before 181.48Mb. None of these genomes contain 181.48Mb-181.54Mb telomere adjacent gap Ns (the black dash arrow in Fig 2), which should be deleted from the hg38 reference. DNA from 4 genomes failed to assemble a contig at this telomere.

**6p**: 69 genomes start near 0.05 Mb of the HG38 reference sequence and the patterns from 0.05Mb to 0.11Mb are significantly different to hg38. 50 genomes start at 0.0 Mb of hg38. Most of the 50 genomes contain additional paralogy block 1–3, and 20 such genomes are shown in Fig 2. DNA from 35 genomes failed to assemble a contig at this telomere.

**6q**: 80 genomes match the hg38 reference. 66 genomes have different variations between 170.74–170.70Mb. One specific pattern containing 15 genomes is shown in Fig 2. All of these haplotypes start near the 170.75 Mb of hg38 reference sequence, and do not contain the 60 kb

telomere-adjacent gap Ns in the HG38 reference sequence. DNA from 8 genomes failed to assemble a contig at this telomere.

**7p**: The 7p subtelomeric region shows a wide range in haplotypes, and all contain extra sequences beyond the hg38 reference. 60 genomes contain 70kb extra sequences, which have the patterns of paralogy blocks of 6-7-8-9. An additional 5 contain 175 kb extra sequences, which comprise of paralogy blocks from 1 to 9, as shown in S1 Fig. 39 genomes contain variable extensions beyond hg38. DNA from 50 genomes did not generate complete assembly due to an inverted nick pair (INP) site, starting at 0.08Mb.

**7q**: 125 genomes match the HG38 reference pattern exactly or have one extra nick site. 17 genomes have 2 additional nick sites and extend 15 kb further than the major haplotype. DNA from the remaining 12 genomes failed to assemble a contig at this telomere.

**8p**: 104 genomes contain the major haplotype that matches the hg38 reference, stopping just before the telomere-adjacent gap Ns in HG38 that begin at 0.06Mb. 38 genomes start at 0.16Mb beginning with block 5 (lacking blocks 24, 25, 28, 19ab, 20 and 4). 11 genomes are the same length as the major haplotype but a different sequence composition for the last 0.09Mb. A set of 3 genomes (sharing one haplotype) out of 11 genomes containing blocks 1, 2, 3 and 4 (but having alternate nicking patterns relative to the set of 3) are shown in S1 Fig.

**9p**: 110 genomes have a haplotype that agrees well with the HG38 reference. 20 genomes have an extended region of 15kb with an unknown block, while 6 genomes have a much longer extended region (60kb of beyond the HG38 reference also unknown block). DNA from the remaining 18 genomes failed to assemble a contig at this telomere.

**9q:** 51 genomes extend 80kb beyond its distal end of hg 38 reference with blocks 3, 4 and some unknown sequence as well. 77 genomes extend into the 60kb telomere adjacent gap Ns of hg38 reference. 15 of these 77 genomes shown in S2 Fig match until 138.3Mb and have an unknown block for their last 30kb region.

**11p**: 11p is highly variable. 41 genomes shown in S2 Fig extend 90kb beyond the telomere-adjacent gap boundary in HG38 reference and contain blocks 1,2,3,4. 80 genomes show variable extension into the 60kb telomere gap Ns. 6 of these 80 extend an additional 80kb past the gap as shown in S2 Fig. 20 genomes among this group extend 0.11Mb from the telomere before matching the HG38 reference pattern and are shown in S2 Fig. DNA from the remaining 33 genomes failed to assemble a contig at this telomere.

**14q**: 95 genomes end at 106.7Mb and do not extend into the telomere gap adjacent Ns. 61 of these genomes have minor differences from the other 34 genomes in the region 106.7Mb to 106.75Mb and in 106.85Mb to 106.86Mb region as seen in S3 Fig. This region is known to contain variable genes from the immunoglobulin G heavy chain cluster [34]. DNA from the remaining 59 genomes failed to assemble a contig at this telomere.

**15q**: 114 genomes match well to the reference. 25 genomes are the same length but have variation from 101.95–101.97Mb. Our previous work [22] revealed what turns out to be a very rare haplotype of 15q with a 50kb extension. Only one genome in the current dataset matched that particular haplotype and it was identical to one from the prior study (GM12892). DNA from 14 genomes failed to assemble a contig at this telomere.

**16q**: 73 genomes match the HG38 reference sequence length and extend only 0.02Mb into the telomere adjacent gap Ns. These match the Stong et al. (2014) 16q assembly. An additional 48 genomes extend around 0.08Mb into the telomere adjacent gap Ns with several variations in this extension. As shown in S3 Fig, 3 of these genomes also differ from the reference at 90.18Mb-90.23Mb and contains blocks 1, 2,3,4,5. Another 5 of these extended genomes match the reference but still include blocks 1, 2, 3, 4. DNA from 33 genomes failed to assemble a contig at this telomere.

**17q**: 105 genomes have a haplotype matching the reference. 19 genomes have a similar pattern but 83.22–83.24 contains variation. S4 Fig shows one example of this group with 3 genomes. DNA from 30 genomes failed to assemble a contig.

**19p**: 62 genomes share a haplotype that matches the end of the reference sequence but is different from that reference at 0.12–0.2Mb, lacking block 5. 44 genomes match the reference and contain block 5. 10 of these 44 genomes extend 0.02Mb into the telomere adjacent gap Ns. DNA from 13 genomes have an 80kb extension with unknown blocks, exceeding the length of the telomere-adjacent gap Ns in the HG38 reference, 3 with the same variation from the reference at 0.12–0.2Mb. S4 Fig shows the 10 that extend and matches block 5 at 0.12Mb. 35 genomes failed to assemble.

**20p:** All of the 135 genomes in 20p extend beyond the 60kb telomere-adjacent gap Ns (black arrow). 94 genomes extend 0.1Mb, 41 genomes extend 0.14Mb (with the distal 0.09Mb differing). DNA from 19 genomes failed to assemble a contig at this telomere.

**Subtelomeres with minimal structural variation.** 18 chromosome arms (1q, 2p, 3p, 4p, 4q, 5p, 8q, 10p, 10q, 11q, 12p, 12q, 13q, 18p, 18q, 20q, 21q, XqYq) show minimal structural variation compared to the HG38 reference. Descriptions and figures depicting these arms can be found in the supplementary materials, S1 Text and S5 Fig to S10 Fig. Among these arms, inconsistencies with the current HG38 reference included 20 kb of extra gap DNA adjacent to the telomere at 8q of the HG38 reference, 110 kb of gap DNA adjacent to 18q of the HG38 reference, and 70 kb of extra gap DNA at 20q of the HG38 reference.

The current HG38 reference does not contain information on the p arms of the acrocentric chromosomes, 22p, 21p, 13p, 14p, and 15p, and thus could not be mapped to our single-molecule assemblies. However, in our recent study, genome maps pick up significant patterns of these acrocentric short-arm subtelomeres [31]. For the XpYp, only a few genomes had contigs matching part of this region. They have similar patterns but variable spacing between the patterns (S6 Fig), and are located centromeric to a large hypervariable pseudoautosomal variable number tandem repeat (VNTR) that has been associated with length polymorphisms up to 100 kb that would be expected to interfere with reference sequence assembly [34, 35].

Four subtelomeres (16p, 17p, 19q and 22q) have known inverted nick pair (INP) sites near the telomere with Nt.BspQ1 labeling, such that the genome maps can't extend to the telomeres. INP sites occur where the nicking enzymes sites are close together on opposing strands, leading to double stranded DNA breaks and thus interference with nick-label mapping.

We further characterized these 4 arms with Direct Label Enzyme (DLE) labeling [36] using a subset of genomes. The DLE enzyme does not nick the DNA, and intact molecules from this analysis confirmed the presence of INP sites using the nicking method. Examples of consensus maps with DLE labeling for these four arms can be found in S11 Fig. Based on this data, 19q and 22q show minimal variations and align well to the hg38 reference. 16p appears to have at least a second longer haplotype as suggested by some of the Nt.BspQ1 genomes, and they also had a longer minor haplotype shown with DLE. The existence of these structurally variant haplotypes is consistent with early PFGE mapping studies showing large subtelomeric polymorphisms at 16p [37]. Genomes labeled with the DLE enzyme showed large differences between mapping patterns for 17p compared with the HG38 reference and further characterization will be needed to accurately classify this arm. Previous 17p subtelomere mapping and sequencing studies showed several discrete large tandem repeat regions within 100 kb of the chromosome end based upon a telomere-containing half-YAC as well as large variations detected by RARE cleavage[30, 38]. Both 17p features may be contributing to difficulties characterizing this subtelomere in the population. The DLE method also mapped very few contigs to arm XpYp, only 10 out of 52 genomes, suggesting, as mentioned above, that the reference may be inaccurate or XpYp may be highly variable with the reference only reflecting one haplotype.

It is clear that the existing human genome reference (HG38) for highly variable subtelomeric regions is often incomplete, especially in the region immediately adjacent to telomere repeats. The uncertainty is often expressed in telomere-adjacent gap regions ranging from 5-160kb on the end of these chromosome arms in the HG38 reference, and these gap sizes correlate poorly with the actual sizes of the structurally variant alleles.

In some cases, we confirmed that extra genomic materials should be added to the end of some arms, such as 2q, 3q, 7p, 9p, 9q, 11p, 19p, and 20p, which can be easily verified by the extra genome map extensions (teal arrow in Fig 2 and S1 Fig to S10 Fig). In other cases, none of the genome maps extend across these telomere-adjacent gaps, which suggests that the regions should be deleted. To further confirm the inaccuracy of specific telomere-adjacent gaps in HG38, we used CRISPR-Cas9 to label the telomere to indicate the end of the genome maps. Fig 3 shows the typical results of this analysis for chromosome arms 1p, 3q, 8p, 14q, and 18q. The raw images of single DNA molecules were shown below the HG38 reference map (blue bar) and consensus genome map (yellow bar). For 8p, the green dots (Nt.BspQ1 motifs) on the blue DNA backbone align really well with the reference map and consensus map. At the end of the molecule, telomeres are shown as a more intense green dot, which was labeled with CRISPR cas9 [32]. Without the CRISPR-cas9 labeling, molecules with the same nick site pattern lack the intense green end labels. This confirms that the approximately 60kb telomere-adjacent gap in the telomere-adjacent region of 8p is inaccurate. A similar conclusion can be drawn for 14q (160kb) and 18q (110kb). As discussed earlier the 1p reference appears inaccurate and contigs start much further toward the centromere. This is confirmed by the presence of the telomere. These extra portions should be deleted from HG38 reference. 3q is an example of an arm that has additional sequences beyond the current reference end.

Table 1 summarizes the detailed comparison between the mapping results and hg38 reference for each chromosome arm. The range in number of genomes per arm is due to some genome assemblies not containing a consensus contig in the distal 500kb region of a chromosome arm. In addition, 5q, 13q, 15q, and 20p have more than 154 genomes, due to some genomes having two contigs (i.e., two long-range subtelomeric haplotypes) for an arm in a diploid genome. Table 1 also includes the current sizes of telomere-adjacent gaps designated in the HG38 reference sequence and the range the contig maps extended beyond that. Differences in telomere-adjacent gap sizes less than 10kb are unable to be distinguished and are estimated as 0. A negative number indicates there is excessive gap size, a positive number indicates insufficient gap size. For 1p and 17p the HG38 reference seems very inaccurate regardless of indicated gap size.

## Population structure of paralogy blocks in subtelomeric regions

We used the consensus Bionano maps to identify subtelomeric paralogy blocks based upon the similarity of their nicking patterns to representative paralogy blocks defined first by Linardopoulou et al. and then extended by Stong et al. [24, 26]. We then explored the population structure of specific subtelomeric paralogy blocks and combinations of adjacent subtelomeric paralogy blocks on each chromosome arm in the 154 genomes at the super-population level; 42 Africans (AFR), 30 Ad Mixed Americans (AMR), 30 East Asians (EAS), 24 Europeans (EUR) and 28 South Asians (SAS) [31].

Paralogy blocks 3, 5 and 9 are each relatively large and have distinct nick-labeling patterns generated by using the nicking enzyme Nt.BspQ1. Some smaller paralogy blocks lack a Nt. BspQ1 nicking site or have only a few, but consistently occur adjacent to and in the same orientation with other small blocks; in these cases we combined paralogy blocks to identify distinctive nicking patterns. Thus, combined blocks 1–2, 6-7-8, and 10-25-11-12 were analyzed as

**Table 1. Summary of chromosomes.**

| Chr Arm | Samples Represented | Large-scale Var[1] | Hg38 Tel-adj Gap[2] (kb) | Map Extension[3] (kb) | Chr Arm | Samples Represented | Large-scale Var[1] | Hg38 Tel-adj Gap[2] (kb) | Map Extension[3] (kb) |
|---|---|---|---|---|---|---|---|---|---|
| 1p | 113 | High | inaccurate | N/A | 1q | 144 | Low | 10 | 0 |
| 2p | 130 | Low | 10 | 0 | 2q | 133 | High | 10 | 0–45 |
| 3p | 143 | Low | 10 | 0 | 3q | 110 | High | 60 | 70–140 |
| 4p | 124 | Low | 10 | 0 | 4q[5] | 133 | N/A | 10 | N/A |
| 5p | 152 | Low | 10 | 0 | 5q | 150 | High | 60 | -60 |
| 6p | 119 | High | 60 | 0 | 6q | 146 | High | 60 | -60 |
| 7p | 104 | High | 10 | 100–175 | 7q | 142 | High | 10 | 0 |
| 8p | 153 | High | 60 | -60 | 8q | 151 | Low | 60 | -20 |
| 9p | 136 | High | 10 | 5–55 | 9q | 128 | High | 60 | -40-20 |
| 10p | 146 | Low | 10 | 0 | 10q[5] | 151 | N/A | 10 | N/A |
| 11p | 121 | High | 60 | 0–80 | 11q | 137 | Low | 10 | -15 |
| 12p | 145 | Low | 10 | 0 | 12q | 147 | Low | 10 | -20 |
| 13p | N/A | N/A | N/A | N/A | 13q | 152 | Low | 10 | 0 |
| 14p | N/A | N/A | N/A | N/A | 14q | 95 | High | 160 | -160 |
| 15p | N/A | N/A | N/A | N/A | 15q | 140 | High | 10 | 0 |
| 16p[4] | 153 | Low | 10 | -10 | 16q | 121 | High | 110 | -20 |
| 17p[4] | 127 | Low | inaccurate | N/A | 17q | 124 | High | 10 | -20 |
| 18p | 144 | Low | 10 | 0 | 18q | 150 | Low | 110 | -110 |
| 19p | 119 | High | 60 | 10 | 19q[4] | 150 | Low | 10 | 0 |
| 20p | 135 | High | 60 | 30–70 | 20q | 148 | Low | 110 | -70 |
| 21p | N/A | N/A | N/A | N/A | 21q | 115 | Low | 10 | 0 |
| 22p | N/A | N/A | N/A | N/A | 22q[4] | 153 | Low | 10 | 0 |
| X/Yp | 33 | Low | 10 | N/A | X/Yq | 139 | Low | 10 | 0 |

Table 1 contains the number of contigs from the 154 genomes present in each arm, the current amount of Ns in the hg38 reference padding, the range of the map extension lengths (if any), and the classification of each arm as High or Low variability. This is determined by looking at the total number of genomes for an arm, and how many were the majority haplotype vs the minor. If the minor haplotype was less than 10% the total, the arm is considered low variability. The acrocentric arms 13p, 14p, 15p, 21p, 22p cannot be determined due to the lack of reference. 4q and 10q have a known repeat D4Z4 in the subtelomeric region and are also excluded [39]. For differences in reference gap and contig length being less than 10kb it is unable to be determined precisely and is estimated as 0. A negative number indicates the gap estimated in the reference is longer than seen in the arm and a positive number indicates the arm is longer than the gap estimated. For 1p and 17p the HG38 reference is very inaccurate.

1—If the minor haplotype was less than 10% the total haplotypes for the arm, the arm is considered low variability.

2—The reference is inaccurate to the point where the size of the telomere adjacent gap of the reference cannot be evaluated

3—For gap differences of <10kb, accuracy is unclear. These arms appear to be within the correct size range. N/A refers to arms that could not accurately be judged, including the acrocentric arms 13p, 14p, 15p, 21p, 22p. In addition, 4q and 10q contain a known D4Z4 repeat leading to widely variable ranges (35).

4—Contain INP sites, and this may affect data that could determine high vs low variability.

5—Can be considered high variability with respect to high levels of large D4Z4 tandem repeat variability in the populations.

three distinctive segments. Fig 4 shows the distribution of paralogy blocks 3, 5, 9, and 1–2 on chromosome arms 15q, 16q and 9q.

Block 3 has an uneven distribution between super populations, with its occurrence being chromosome- and population-dependent. S1 Table contains a detailed breakdown of the frequency of this block. Chromosome arms 3q, 6q, 15q, and 19p show a higher prevalence of block 3 in the majority of people with similar frequencies among super populations. Chromosome arms 2q, 5q, 6p, 7p, 8p, 9q, and 16q have lower prevalence of block 3 among all super populations. Arm 16q is statistically (p<0.05, Bonferroni correction) higher prevalence in the African super population compared to the other 4 super populations. This confirms the

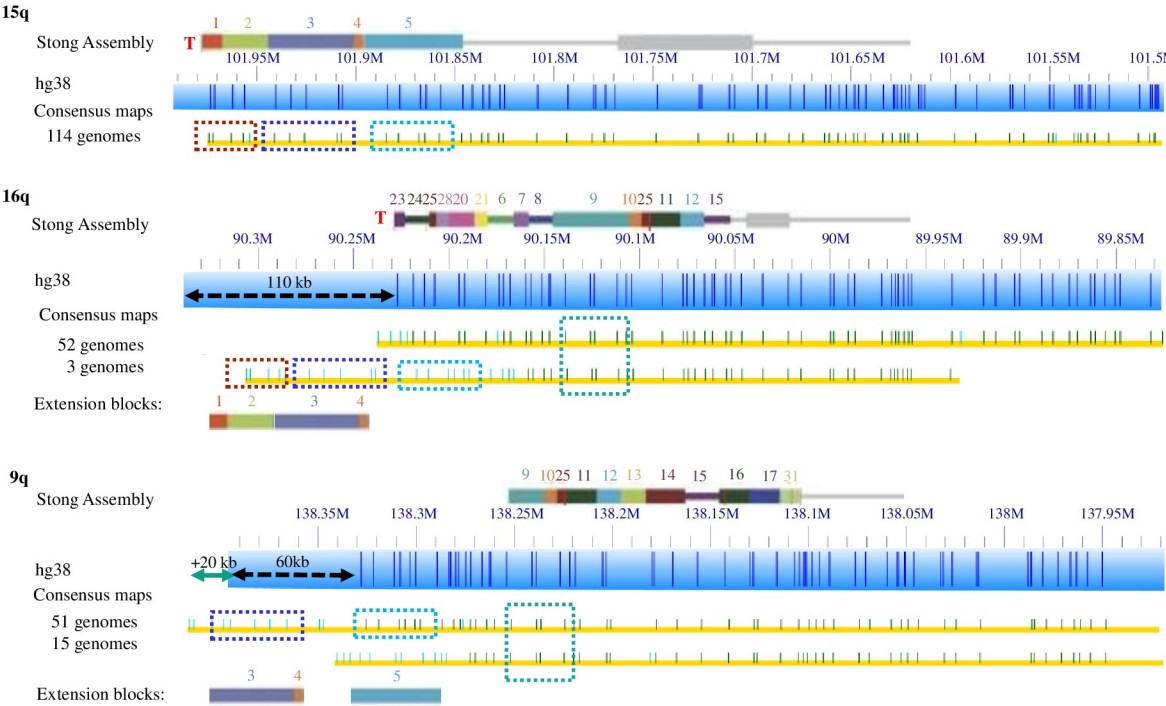

**Fig 4. Distribution of paralogy block 5 in 15q, 16q and 9q.** The solid color rectangle bars show the paralogy blocks defined in the subtelomeric assemblies of Stong et al. (2014). The narrow grey line segments to the right of the colored blocks show the single-copy DNA region. Blue rectangles with dark blue lines show the HG38 reference with Nt.BsPQ1 nick sites. Paralogy block five is shown as a dashed blue rectangle on top of yellow rows representing consensus maps for particular genomes. Additional paralogy blocks are also shown as dashed colored rectangles. A teal arrow indicates the size of additional extended regions not covered by the reference. A black arrow represents unknown sequence currently in the HG38 reference as 'N', an estimate of gap size to the end of the chromosome. If the black arrow is dashed it signifies a region of unknown telomere-adjacent gap sequence that should be deleted.

previous analysis of cosmid f7501 (with DNA sequence nearly identical to a portion of block 2 and half of block 3) by Trask et al, where fluorescence in situ hybridization localized the sequence on almost all genomes for 3q, 15q, 19p and only on a few genomes for 7p from African pygmy tribes [40]. An additional analysis of cosmid f7501 by Der Sarkissian et al also found it on these arms as well[41]. However, this study also found cosmid f7501 on 5q for one sample. We did not find any other copy on chromosome 5q from these 154 samples. This is consistent with Der Sarkissian study that one copy was found from a set of 62 Caucasian samples[41]. Block 3 is commonly found with Block 1 and 2. Block 1 and 2 show similar trends as block 3 on arm 16q, where block 1–2 is significantly more common in the African super population than the other super populations. Block 1–2 is found in just 3 genomes for chromosome arm 8p, all belonging to the African super population (S1 Table), but it is not statistically different than other super populations.

Block 5 also has an uneven distribution between super populations (Fig 4). Block 5 has higher prevalence on chromosome arms 3q, 5q, 6p, 6q, 8p, 11p, and 15q with similar distributions between super populations (Table 2). However, block 5 has higher distribution on chromosome arms 9q and 16q in the AFR super population. Our results support the findings of previous studies of this paralogy block that could signify recent human divergence [40]. Block 5 may have only spread to 9q and 16q in African populations after the other ancestral populations left Africa, resulting in its appearance there primarily in African populations. Alternatively, it may have spread to these arms prior to the divergence but became reduced in frequency in non-African populations due to genetic drift in those populations that left.

**Table 2. Distribution of block 5 by population.**

| Block 5 | AFR with Block 5 / Total AFR maps | AMR with Block 5 / Total AMR maps | EAS with Block 5 / Total EAS maps | EUR with Block 5 / Total EUR maps | SAS with Block 5 / Total SAS maps |
|---|---|---|---|---|---|
| 2q | 0% | 0% | 0% | 0% | 4% |
| 3q | 54% | 61% | 76% | 65% | 35% |
| 5q | 88% | 90% | 97% | 92% | 82% |
| 6p | 56% | 41% | 21% | 22% | 32% |
| 6q | 83% | 77% | 62% | 78% | 80% |
| 7p | 7% | 0% | 0% | 3% | 7% |
| 8p | 81% | 81% | 67% | 88% | 78% |
| 9q | 62% | 30% | 48% | 32% | 15% |
| 11p | 49% | 42% | 21% | 43% | 35% |
| 15q | 93% | 97% | 97% | 92% | 93% |
| 16q | 14% | 0% | 0% | 0% | 4% |
| 19p | 10% | 27% | 47% | 18% | 19% |

Table 2 shows the frequency of Block 5 on different chromosomes (rows) for each super population (columns). AFR stands for the Africa super population category, AMR for Ad Mixed American, EAS for East Asian, EUR for European, and SAS for South Asian. Based on the Stong reference block 5 had previously been found on 5q, 6p, 6q, 8p, 11p, 15q, and 19p. In our dataset it was also found on 2q, 3q, 7p, 9q, and 16q.

Blocks group 9, group 6-7-8, and group 10-25-11-12 do not show any statistically significant differences in frequencies between super populations.

Arms including 3q, 2q, 9p, 9q, 11p, 15q, 17p, and 19p have additional extended regions compared to the HG38 reference. These extended regions don't belong to any known paralogy blocks. These could represent new combinations of existing subtelomeric segmental duplication material, or subtelomeric insertions of material from elsewhere in the genome. Most of the arms show even distributions of the extended regions between super populations, except arms 17p and 9p. These two arms have extra extensions only in AFR super population, but with relatively low frequencies. Block 19a is related to probe DNF92. A previous study by Der Sarkissian et al of this probe showed more locations than seen by our block 19a. Specifically, we found this block19a on arms 1p, 5q, 6q, 8p and 17q, but not on arms 7p, 9q, 11p, 15q, 16p, 16q. However, arms of 7p, 9q,11p,15q,16p and 16q may still contain a small portion of block 19a, which is not detected by optical mapping [41].

At the super population level, in general, paralogy blocks are present at equal frequency on all five super-populations on most of the chromosome arms that contain the paralogy blocks. However, a few chromosome arms show significantly different frequencies of paralogy blocks (blocks 1–5 and additional extended regions beyond the HG38 reference) in the African super-population. These paralogy blocks are DNA segments immediately adjacent to the telomere, which may be associated with their relatively rapid duplication and spread amongst multiple human subtelomeres.

## Discussion

In this study we utilized optical mapping [27, 42] for 154 individual genomes. We used long DNA molecules (>150kb) and a minimum coverage depth of 60x for each genome. For each genome, every chromosome arm was compared with the hg38 reference and the Stong

subtelomeric reference [24]. The current HG38 reference contains telomere-adjacent gaps represented by strings of "N"s corresponding to missing base pairs in the reference. Our data shows that these telomere-adjacent gaps are frequently inaccurate and represent both structural variations as well as sequence gaps. Like our previous 6 sample study, we were able to detect new long range haplotypes [22]; here we created a much more comprehensive catalog of these alternative haplotypes and their relative frequencies in different populations. 18 chromosome arms (1p, 2q, 3q, 5q, 6p, 6q, 7p, 7q, 8p, 9p, 9q, 11p, 14q, 15q, 16q, 17q, 19p, 20p) are classified as highly variable (minor haplotypes comprised more than 10% of the total genomes). 18 other chromosome arms (1q, 2p, 3p, 4p, 4q, 5p, 8q, 10p, 10q, 11q, 12p, 12q, 13q, 18p, 18q, 20q, 21q, XqYq) showed minimal variations compared to the hg38 reference sequences.

Specific subtelomeres from some genomes failed to assemble and/or map to the reference genome. The reason for this is unclear. These failures may be due to problems of short molecule lengths or low labeling density causing samples to form shorter contigs than normal. The individual genomes missing subtelomere assemblies were not consistent, precluding non-specific DNA fragmentation in certain samples as the cause for these failures to assemble.

Using these mapping datasets, we also examined the distribution of the Stong reference paralogy blocks between the 5 super populations. Block group 1–2, block 3, and block 5 showed a statistically significant prevalence on chromosome arm 16q, in the AFR super population over the other super populations. There is not a significant difference on other arms or in the other blocks.

Some genomes had haplotypes with extension beyond the hg38 reference which did not match any known block in the Stong reference. These also seemed to be more prevalent AFR super population, as the others did not contain the unknown block extension. However, this novel extension in AFR was not statistically significant, possibly due to the small sample size.

From this dataset, we can speculate on the timeline of the development of the paralogy blocks in subtelomeric regions based on their distribution between super populations. In the case of block 3, it is very common on 3 arms, rare on 3 other chromosomes and not found on the rest. Chromosome arms 3q, 15q, and 19p show heavy prevalence of block 3 with a majority of people in all populations having the block, so block 3 may have spread to these arms first and later in 3 chromosome arms (7p, 16p, 16q) where only a few genomes have block 3. Only 16q was significantly different than the other blocks. These distribution differences could be attributed to human migration out of Africa. The development of block 3 on 3q, 15q and 19p likely predates the migration and subsequent expansion of human populations out of Africa [43]. 7p, 16p and 16q potentially developed their instance of block 3 after the initial migration, and spread to all populations remaining in Africa. Alternatively, it may have existed but not yet become a fixed allele at the time of migration and lost over time to genetic drift, and thus not be found in non-African populations. However, arms 2q, 5q, 6p, and 8p do not share this segregation, as their block 3 is present in small frequencies (<10%) and evenly distributed among super populations. The reasons for this even distribution are unknown. It could be that the mechanisms of subtelomeric diversity have led to each of these populations simultaneously developing them independently. The exact mechanisms of subtelomeric variation remains undetermined. Overall, these results support the findings of a previous study of paralogy block 3 that used 8 isolated ethnic groups (2 Pygmy groups, Melanesian, 2 Amerindian, Khmer, Druze and Caucasian, 45 total genomes) and 8 primate genomes, that signifies a recent human divergence [40].

Other paralogy block groups, such as block group 6-7-8, showed different trends, where a majority of all populations had the block and there were no statistically significant difference between super populations for any arm. These blocks are likely to have developed and spread to the arms earlier than the unevenly distributed blocks.

Subtelomeric segmental duplications (often referred to as subtelomeric repeats) are an almost universal feature of eukaryotic genomes and, like in humans, are highly variable and nearly impossible to assemble with current sequencing methods. In addition, the expansion and rapid evolution of functional gene families associated with subtelomeres has been noted in a very wide range of species, from yeasts and protozoans to complex plants and animals [14, 44–46]. Application of this methodology could be used to efficiently map large-scale subtelomeric structural variation in any eukaryote to enable completion of accurate subtelomeric assemblies and analysis of embedded gene families.

This work catalogs a large number of novel long-range subtelomere haplotypes and determines their frequencies and contexts in terms of specific subtelomeric duplicons on each chromosome arm. This information will provide mapping guideposts for their eventual sequence determination, and helps to clarify the currently ambiguous nature of many specific subtelomere structures as represented in the current reference sequence (HG38). As such, this information is an essential step in understanding the impact of subtelomeric cis-sequences on transcription of TERRA and other functional subtelomeric RNAs and their roles in regulating single-telomere lengths and function, as well as delineating the population structure and evolution of highly variable human subtelomere regions.

## Methods

### High molecular weight DNA extraction

Mammalian cells were embedded in gel plugs and High Molecular Weight DNA was purified as described in a commercial large DNA purification kit (BioRad #170–3592). Plugs were incubated with lysis buffer and proteinase K for four hours at 50°C. The plugs were washed and then solubilized with GELase (Epicentre). The purified DNA was subjected to four hours of drop-dialysis. It was quantified using Quant-iTdsDNA Assay Kit (Life Technology), and the quality was assessed using pulsed-field gel electrophoresis.

### DNA labeling

The DNA was labeled with nick-labeling [47] as described previously using the IrysPrep Reagent Kit (BioNano Genomics). Specifically, 300 ng of purified genomic DNA was nicked with 7 U nicking endonuclease Nt.BspQI (New England BioLabs, NEB) at 37°C for two hours in NEB Buffer 3.1. The nicked DNA was labeled with a fluorescent-dUTP nucleotide analog using Taq polymerase (NEB) for one hour at 72°C. After labeling, the nicks were ligated with Taq ligase (NEB) in the presence of dNTPs. The backbone of fluorescently labeled DNA was stained with YOYO-1 (Invitrogen).

A subset of the samples were labeled using the newer Direct Label Enzyme (DLE) method (Bionano Genomics). These samples used the DNA Labeling Kit-DLS 80005 and followed the manufacturer's instructions. In Summary, 750ng of the gDNA was labeled using DLE-1 enzyme and reaction mix followed by Proteinase K digestion (Qiagen). The DNA back bone was stained after drop dialysis. The stained sample was then homogenized and incubated at room temperature over nigh before quantified using Qubit dsDNA HS Kit (Invitrogen).

### Data collection

The DNA was loaded onto the nano-channel array of BioNano Genomics IrysChip by electrophoresis of DNA. Linearized DNA molecules were imaged using a custom made whole genome mapping system. The DNA backbone (outlined by YOYO-1 staining) and locations of fluorescent labels along each molecule were detected using an in-house image detection

software. The set of label locations relative to the DNA backbone for each DNA molecule defines an individual single-molecule map. A commercial version of this whole-genome mapping and imaging system (Irys) is available from Bionano Genomics.

## De novo genome map assembly

Single-molecule maps were assembled *de novo* into consensus maps using software tools developed at BioNano Genomics, specifically Refaligner and Assembler [1]. Briefly, the assembler is a custom implementation of the overlap-layout-consensus paradigm with a maximum likelihood model. An overlap graph was generated based on pairwise comparison of all molecules as input. Redundant and spurious edges were removed. The assembler outputs the longest path in the graph and consensus maps were derived. Consensus maps are further refined by mapping single molecule maps to the consensus maps and label positions are recalculated. Refined consensus maps are extended by mapping single molecules to the ends of the consensus and calculating label positions beyond the initial maps. After merging of overlapping maps, a final set of consensus maps was output and used for subsequent analysis. The map assemblies are very robust to the relatively small errors in labeling (10% false positive, due to extra nickings at wrong sites, and 10% false negative, due to missing nicks). This does not affect the maps and haplotype calls as the haplotypes are both are based on multiple nicking sites and multiple single molecules.

## Block definition

Subtelomeric paralogy blocks originally defined by Linardopoulou et al and extended/refined by Stong et al., [24, 26] are sequence segments of highly similar duplicated subtelomeric DNA that can be identified as discrete contiguous duplicated DNA segments in subtelomere reference assemblies[24]. Paralogy blocks were characterized for mapping purposes by the pattern of nick sites in the representative sequenced reference paralogy block or set of adjacent paralogy blocks. These nicking patterns were then compared with the subtelomere regions of maps generated for each genome. Block boundaries were identified by a qualitative comparison based on the distance between nick sites and their pattern on several arms with shared blocks.

## Statistical analysis of paralogy blocks in super population

An analysis of variant (ANOVA) was calculated on the block presence per genome, grouped by super population. A Bonferroni correction was then performed to determine significance between the 4 super populations. This statistical analysis was repeated independently for each paralogy block or block group analyzed.

## Supporting information

**S1 Fig. Major haplotypes of highly variable subtelomere regions additional chromosome arms 7p, 7q and 8p.** S1 Fig shows the major haplotypes for additional chromosome arms 7p, 7q and 8p in the highly variable set of subtelomeres.). The Stong Assembly paralogy blocks are shown as colored rectangles above blue Bionano optical mapping bars. Yellow rows with green ticks show haplotypes below these. A teal arrow indicates the size of additional extended regions not covered by the reference. A black arrow indicates the region indicated as a telomere-adjacent gap in the HG38 reference sequence. If the black arrow is dashed it signifies a region that should be deleted.
(TIF)

**S2 Fig. Major haplotypes of highly variable subtelomere regions additional chromosome arms 9p, 9q and 11p.** Chromosome arms 9p, 9q and 11p are shown. The Stong Assembly paralogy blocks are shown as colored rectangles above blue Bionano optical mapping bars. Yellow rows with green ticks show haplotypes below these. A teal arrow indicates the size of additional extended regions not covered by the reference. A black arrow indicates the region indicated as a telomere-adjacent gap in the HG38 reference sequence. If the black arrow is dashed it signifies a region that should be deleted.
(TIF)

**S3 Fig. Major haplotypes of highly variable subtelomere regions additional chromosome arms 14q, 15q and 16q.** Chromosome arms 14q, 15q and 16q are shown. The Stong Assembly paralogy blocks are shown as colored rectangles above blue Bionano optical mapping bars. Yellow rows with green ticks show haplotypes below these. A teal arrow indicates the size of additional extended regions not covered by the reference. A black arrow indicates the region indicated as a telomere-adjacent gap in the HG38 reference sequence. If the black arrow is dashed it signifies a region that should be deleted.
(TIF)

**S4 Fig. Major haplotypes of highly variable subtelomere regions additional chromosome arms 17q, 19p, and 20p.** Chromosome arms 17q, 19p and 20p are shown. The Stong Assembly paralogy blocks are shown as colored rectangles above blue Bionano optical mapping bars. Yellow rows with green ticks show haplotypes below these. A teal arrow indicates the size of additional extended regions not covered by the reference. A black arrow indicates the region indicated as a telomere-adjacent gap in the HG38 reference sequence. If the black arrow is dashed it signifies a region that should be deleted.
(TIF)

**S5 Fig. Major haplotypes of less variable subtelomere regions shown for chromosome arms 1q, 2p, and 3p.** S5 Fig shows the major haplotypes for chromosome arms 1q, 2p and 3p in the less variable set of subtelomeres. The Stong Assembly paralogy blocks are shown as colored rectangles above blue Bionano optical mapping bars. Yellow rows with green ticks show haplotypes below these. A teal arrow indicates the size of additional extended regions not covered by the reference. A black arrow indicates the region indicated as a telomere-adjacent gap in the HG38 reference sequence. If the black arrow is dashed it signifies a region that should be deleted. Each low-variability arm (S5-S10) is briefly described in S1 Text.
(TIF)

**S6 Fig. Major haplotypes of less variable subtelomere regions shown for chromosome arms 4p, 4q and 5p.** S6 Fig shows the major haplotypes for chromosome arms 4p, 4q and 5p in the less variable set of subtelomeres. The Stong Assembly paralogy blocks are shown as colored rectangles above blue Bionano optical mapping bars. Yellow rows with green ticks show haplotypes below these. A teal arrow indicates the size of additional extended regions not covered by the reference. A black arrow indicates the region indicated as a telomere-adjacent gap in the HG38 reference sequence. If the black arrow is dashed it signifies a region that should be deleted.
(TIF)

**S7 Fig. Major haplotypes of less variable subtelomere regions shown for chromosome arms 8q, 10p and 10q.** S7 Fig shows the major haplotypes for chromosome arms 8q, 10p and 10q in the less variable set of subtelomeres. The Stong Assembly paralogy blocks are shown as colored rectangles above blue Bionano optical mapping bars. Yellow rows with green ticks show

haplotypes below these. A teal arrow indicates the size of additional extended regions not covered by the reference. A black arrow indicates the region indicated as a telomere-adjacent gap in the HG38 reference sequence. If the black arrow is dashed it signifies a region that should be deleted.
(TIF)

**S8 Fig. Major haplotypes of less variable subtelomere regions shown for chromosome arms 11q, 12p, and 12q.** S8 Fig shows the major haplotypes for chromosome arms 11q, 12p, and 12q in the less variable set of subtelomeres. The Stong Assembly paralogy blocks are shown as colored rectangles above blue Bionano optical mapping bars. Yellow rows with green ticks show haplotypes below these. A teal arrow indicates the size of additional extended regions not covered by the reference. A black arrow indicates the region indicated as a telomere-adjacent gap in the HG38 reference sequence. If the black arrow is dashed it signifies a region that should be deleted.
(TIF)

**S9 Fig. Major haplotypes of less variable subtelomere regions shown for chromosome arms 13q, 18p, and 18q.** S9 Fig shows the major haplotypes for chromosome arms 13q, 18p, and 18q in the less variable set of subtelomeres. The Stong Assembly paralogy blocks are shown as colored rectangles above blue Bionano optical mapping bars. Yellow rows with green ticks show haplotypes below these. A teal arrow indicates the size of additional extended regions not covered by the reference. A black arrow indicates the region indicated as a telomere-adjacent gap in the HG38 reference sequence. If the black arrow is dashed it signifies a region that should be deleted.
(TIF)

**S10 Fig. Major haplotypes of less variable subtelomere regions shown for chromosome arms 20q, 21q, XpYp and XqYq.** S10 Fig shows the major haplotypes for chromosome arms 20q, 21q, XpYp and XqYq in the less variable set of subtelomeres. The Stong Assembly paralogy blocks are shown as colored rectangles above blue Bionano optical mapping bars. Yellow rows with green ticks show haplotypes below these. A teal arrow indicates the size of additional extended regions not covered by the reference. A black arrow indicates the region indicated as a telomere-adjacent gap in the HG38 reference sequence. If the black arrow is dashed it signifies a region that should be deleted.
(TIF)

**S11 Fig. Direct Label Enzyme (DLE) labeling HG38 reference and contigs for arms 16p, 17p, 19q, 22q.** Reference and consensus maps for arms 16p, 17p, 19q, and 22q are shown. Blue bars with dark blue lines indicate the location of labels in HG38. Yellow rows with green ticks show haplotypes below these for each arm.
(TIF)

**S1 Table. Block Frequencies by chromosome and superpopulation.** S1 Table shows the frequency of Block 1–2, Block 3, Block 6-7-8, Block 9, and Blocks 10-11-25-12 on different chromosomes (rows) for each super population (column). AFR stands for the Africa super population category, AMR for Ad Mixed American, EAS for East Asian, EUR for European, and SAS for South Asian.
(XLSX)

**S1 Text. Descriptions of haplotypes found in low variability arms.**
(DOCX)

## Acknowledgments

Part of the informatics analysis was run on hardware supported by Drexel's University Research Computing Facility. We thank Bionano Genomics Inc. for assistance in data generation and bioinformatics support.

## Author Contributions

**Conceptualization:** Pui-Yan Kwok, Harold Riethman, Ming Xiao.

**Data curation:** Eleanor Young, Pui-Yan Kwok, Ming Xiao.

**Formal analysis:** Eleanor Young, Heba Z. Abid, Harold Riethman, Ming Xiao.

**Funding acquisition:** Eleanor Young, Pui-Yan Kwok, Harold Riethman, Ming Xiao.

**Methodology:** Eleanor Young, Ming Xiao.

**Project administration:** Harold Riethman, Ming Xiao.

**Software:** Eleanor Young.

**Supervision:** Ming Xiao.

**Validation:** Eleanor Young, Harold Riethman, Ming Xiao.

**Visualization:** Eleanor Young, Harold Riethman, Ming Xiao.

**Writing – original draft:** Eleanor Young, Ming Xiao.

**Writing – review & editing:** Eleanor Young, Pui-Yan Kwok, Harold Riethman, Ming Xiao.

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
