## [Decision Letter · Decision Letter 0]

9 Sep 2019

Dear Dr Xiao,

Thank you very much for submitting your Research Article entitled 'Comprehensive Analysis of Human Subtelomeres by Whole Genome Mapping' to PLOS Genetics. Your manuscript was fully evaluated at the editorial level and by independent peer reviewers. The reviewers appreciated the attention to an important topic but identified some aspects of the manuscript that should be improved.

We therefore ask you to modify the manuscript according to the review recommendations before we can consider your manuscript for acceptance. Your revisions should address the specific points made by each reviewer.

[LINK]

Yours sincerely,

Nancy Maizels, Ph.D.

Associate Editor

PLOS Genetics

Scott Williams

Section Editor: Natural Variation

PLOS Genetics

Reviewer's Responses to Questions

**Comments to the Authors:**

Reviewer #1: The manuscript by Young et al describes the subtelomeric structure and variation across human populations using optical mapping of large single DNA molecules. This is a powerful technique allowing to assemble and connect subtelomeric regions (blocks) repeated throughout the genome to specific chromosome sequences, thus precisely revealing the structure of almost all proterminal regions of human chromosomes. The authors analyzed data collected from 154 human genomes representing 26 populations around the world. Genome wide maps using the same data have been described and published recently by the same group in another journal. In this manuscript, the analyses concentrate on subterminal regions known for longtime to be highly polymorphic to the point that they are excluded most of the time from the consensus human genome sequence. The catalog provides with unprecedented detail a description of the structure and variation of the most frequent haplotypes at many subtelomeric domains. Although highly descriptive, this study perfectly illustrates the extent of variation of such regions as well as the potential of long-range optical mapping to complete and correct the current assembly of the human genome.

Some issues, though, must be resolved before the manuscript can be accepted for publication:

Major points:

A previous description of subtelomeric dynamics using FISH probes (Der Sarkissian et al, Genome Res 2002) that partially overlap the subtelomeric blocks as defined later by the Trask and subsequently by the Riethman laboratories had already given an idea of the structure and extent of subtelomeric poplymorphisms in human populations, in particular at specific chromosome ends. For instance, the Der Sarkissian study already describes an organization of the 3q telomere that connects sequences in the cosmid f7501 (block 2 and 3, first described by Trask) to the blocks described in the Stong assembly. This should be mentioned and the corresponding reference cited.

Also, even if the number of genomes analyzed in this work is much larger and more diverse than previous studies using FISH, it fails to capture all the variability originally described in Der Sarkissian et al. For instance, the block 19a defined by Stong is a major and invariable component of the 17q subtelomere as shown here. This block is likely included (at least partially) in the DNF92 probe used in the Der Sarkissian study, which also found 17q to be invariable. However, that study also found that DNF92 is present at many more extremities (3q, 6p, 7p, 9q, 11p, 15q, 16p, 16q) while in the present study it is restricted to a few highly variable extremities. On the other hand, the Der Sarkissian study also shows that the cosmid f7501 (blocks 2/3) can also be present at 5q, a variation absent from the alleles described here. While some of these discrepancies, which should be discussed in the text, may be explained by technical issues, this suggests that variation is even more important than anticipated and that more genomes need to be analyzed by optical mapping in order to get a more precise idea of the prevalence of such variations.

Other points:

Proofreading is required throughout the text as well as in the reference list.

Reviewer #2: In this manuscript Young et. al. attempt to use whole genome mapping to characterize the subtelomeric region that has been lacking in the reference genome HG38. They investigated the human subtelomere structure and variation in 154 human genomes across 26 populations and found that 18 chromosome arms were highly variable. They found that the major haplotype for a number of chromosome ends differed to the reference genome HG38, either extending beyond the region of known sequence or missing sections of the terminal end of the reference. The authors also investigate the distribution of the Stong reference paralogy blocks between the 5 super populations, finding that block groups 1-2, block 3, and block 5 had a significantly high prevalence on chromosome arm 16q in the African super population compared to the remaining populations.

Overall the findings in this manuscript provide new insight into how much the subtelomeres vary across the human population and how they differ to the reference genome HG38. This provides an important consideration when attempting to study the subtelomere/telomere regions using sequencing by mapping to the reference genome.

Major comments:

• The major contig of the chromosomes (eg. 5q, 6p) don’t span the entire region shown for hg38. Is this indicative that the proximal end of the region shown is not present in these genomes? Are these large deletion events or are these contigs not attached to the rest of the chromosome?

• Results section on 11p: the 6 genomes with a 140kb extension from the telomere adjacent gap boundary are shown in Figure 2D but not mentioned in the text.

• Results for experiments using DLE not shown. Data need to be shown and referenced for results paragraph starting at line 259.

• Line 356: the authors state “Arm 16q is statistically (p<0.05, Bonferroni correction) lower in the African super population compared to the other 4 super populations.”, are you referring to the presence of Block 3? If so, then is it not meant to be higher as Table S1 shows? Also, it would be better to refer to Table S1 at the start of this paragraph before discussing the results.

• The text regarding 16q does not match Figure 2E. Line 220 states “73 genomes” while the figure mentions “52” match the HG38 reference. Which is correct?

Minor comments:

• Missing label for F panel in Figure 2.

• Typo line 139: “sequence content of sequence content of” should be “sequence content of”.

• Type line 177: 2 extra ‘.’ at the end of the sentence.

• “INP” not defined on line 188.

• Figure 2E, for 15q the figure states 26 contigs while the text states 25 contigs (line 215).

• Typo Line 220: “HG 38” should be “HG38”.

• “VNTR” not defined on line 253.

• “DLE” not defined in first use on line 259.

• Typo line 279: line ends with multiple periods.

• Typo line 303: “Table one” should be “Table 1”.

Reviewer #3: The subtelomeric regions of eukaryotic genomes have been a black box in almost all genome projects due to their structure and dynamic nature, exhibiting presence/absence, CNV and location variation amongst chromosome ends, individuals and populations. Even in small genomes the subtelomeric regions sharing blocks of homology across more than one chromosome end can span 10s to 100s of kilobases, which are generally hemizygous, making it difficult to map any sequence reads to specific locations. In first generation genome projects the ends of chromosomes were underrepresented in libraries, requiring specialised efforts to enrich for large telomere containing clones or to label specific telomeres for targeting cloning/sequencing. In second generation sequencing methodology the short reads cannot resolve the issues of duplications and haplotype phasing required to assemble subtelomeres. Long read third generation methods may eventually solve the subtelomere assembly and characterisation problem but at the moment the read lengths, accuracy and throughput are not sufficient for high throughput effective characterisation of large numbers of individual genomes. The issue has been exemplified by the human subtelomeres, where one of the authors, Harold Riethman, has worked tirelessly over 30 years to characterise these regions of the human genome. First using large clones enriched for telomeres/subtelomeres and Sanger sequencing and then more recently using improved informatics and mixed sequencing platforms which was a great improvement over the original human genome project efforts. This manuscript reports a significant advance in the characterising of all subtelomeres in a large sample of individuals from several populations in the 1000 human genome project.

Specific physical mapping of sequence tagged nicks on long single DNA molecules allows mapping known subtelomeric repeat elements from the unique sequences of each chromosome end to the telomere, filling gaps in the contigs from the hunan genome reference and the author's more complete characterisation in 2014. When anchoring nicks were too close resulting in DSBs an alternative method was used resulting in the characterisation of all subtelomeres in 6 individuals each from 26 different populations. A tour de force the will stand as the definitive study until 3rd generation sequencing becomes more accurate over long enough ranges at a cost and efficiency that can handle the numbers.

Some subtelomeres are highly variable in composition and copy number of different STEs while others are more homogeneous. Some inference can be made about population genetics/evolution of subtelomeres and some STEs but the sample sizes within each population may not be large enough to conclude much about origins and spread.

One potential issue is the cells lines used to isolate the DNA from. Are they genomically stable or is there some instability arising from the establishment of the cell lines which may have resulted in alterations of subtelomeres due to recombinational dynamics? Given that many subtelomeres are homogeneous in the sample argues against instability in the cell lines though some ends may be more unstable than others.

Overall I am very impressed with the effort and presentation. It would be nice to mention other organisms where this methodology would be useful - ie where subtelomeres are known to be important for many phenotypes.

**Have all data underlying the figures and results presented in the manuscript been provided?**

Reviewer #1: No:

Reviewer #2: Yes

Reviewer #3: Yes

PLOS authors have the option to publish the peer review history of their article (what does this mean?). If published, this will include your full peer review and any attached files.

Reviewer #1: Yes: Arturo Londono-Vallejo

Reviewer #2: No

Reviewer #3: No

---

## [Editor Report · Decision Letter 1]

15 Oct 2019

Dear Dr Xiao,

We are pleased to inform you that your manuscript entitled "Comprehensive Analysis of Human Subtelomeres by Whole Genome Mapping" has been editorially accepted for publication in PLOS Genetics. Congratulations!

Yours sincerely,

Nancy Maizels, Ph.D.

Associate Editor

PLOS Genetics

Scott Williams

Section Editor: Natural Variation

PLOS Genetics

Comments from the reviewers (if applicable):

**Data Deposition**

http://datadryad.org/submit?journalID=pgenetics&manu=PGENETICS-D-19-01274R1

Press Queries

---

## [Editor Report · Acceptance letter]

9 Jan 2020

PGENETICS-D-19-01274R1 

Comprehensive Analysis of Human Subtelomeres by Whole Genome Mapping 

Dear Dr Xiao, 

We are pleased to inform you that your manuscript entitled "Comprehensive Analysis of Human Subtelomeres by Whole Genome Mapping" has been formally accepted for publication in PLOS Genetics! Your manuscript is now with our production department and you will be notified of the publication date in due course.

With kind regards,

Matt Lyles

PLOS Genetics

On behalf of:
